# Mapping the probability of forest snow disturbances in Finland

Susanne Suvanto[1,2]*, Aleksi Lehtonen[1], Seppo Nevalainen[3], Ilari Lehtonen[4], Heli Viiri[5], Mikael Strandström[1], Mikko Peltoniemi[1]

**1** Natural Resources Institute Finland (Luke), Helsinki, Finland, **2** University of Birmingham, School of Geography, Earth and Environmental Sciences, Birmingham, United Kingdom, **3** Natural Resources Institute Finland (Luke), Joensuu, Finland, **4** Finnish Meteorological Institute, Helsinki, Finland, **5** UPM Forest, Tampere, Finland

* susanne.suvanto@luke.fi

## Abstract

The changing forest disturbance regimes emphasize the need for improved damage risk information. Here, our aim was to (1) improve the current understanding of snow damage risks by assessing the importance of abiotic factors, particularly the modelled snow load on trees, versus forest properties in predicting the probability of snow damage, (2) produce a snow damage probability map for Finland. We also compared the results for winters with typical snow load conditions and a winter with exceptionally heavy snow loads. To do this, we used damage observations from the Finnish national forest inventory (NFI) to create a statistical snow damage occurrence model, spatial data layers from different sources to use the model to predict the damage probability for the whole country in 16 x 16 m resolution. Snow damage reports from forest owners were used for testing the final map. Our results showed that best results were obtained when both abiotic and forest variables were included in the model. However, in the case of the high snow load winter, the model with only abiotic predictors performed nearly as well as the full model and the ability of the models to identify the snow damaged stands was higher than in other years. The results showed patterns of forest adaptation to high snow loads, as spruce stands in the north were less susceptible to damage than in southern areas and long-term snow load reduced the damage probability. The model and the derived wall-to-wall map were able to discriminate damage from no-damage cases on a good level (AUC > 0.7). The damage probability mapping approach identifies the drivers of snow disturbances across forest landscapes and can be used to spatially estimate the current and future disturbance probabilities in forests, informing practical forestry and decision-making and supporting the adaptation to the changing disturbance regimes.

## Introduction

Forest disturbances caused by snow are frequent in high latitude regions [1–4] and high-altitude areas [5–7]. In Europe, the estimates of forest damage caused by snow disturbance events

**Data Availability Statement:** The NFI field data was used here under a permission from the National Forest Inventory of Finland (within Natural Resources Institute Finland) that does not allow the

authors to publicly share this data. The MS-NFI data is available for download under CC-BY 4.0 license from http://kartta.luke.fi/index-en.html, the data for snow load on tree crowns is available under CC-BY 4.0 license from http://urn.fi/urn:nbn: fi:csc-kata20180329115425634000, the DEMs are available under CC-BY 4.0 license from https:// tiedostopalvelu.maanmittauslaitos.fi/tp/kartta? lang=en, the data on biogeographical regions in Finland are available under CC-BY 4.0 license from https://ckan.ymparisto.fi/dataset/%7B664BE696- C6A5-4FC4-8D6A-7D2E63D0E9C6%7D. The data from Finnish Forest Centre can be downloaded under CC-BY 4.0 license from https://www. metsaan.fi/paikkatietoaineistot.

**Funding:** The research was funded from the project SÄÄTYÖ funded by the Ministry of Agriculture and Forestry of Finland. This project has received funding from the European Union's Horizon 2020 research and innovation programme under the Marie Skłodowska-Curie grant agreement No 895158. The contribution of Mikko Peltoniemi on the research was also funded through the BiodivClim ERANet Cofund (joint BiodivERsA Call on "Biodiversity and Climate Change", 2019-2020) with national co–funding through Academy of Finland (decision no. 344722), ANR (France, project ANR-20-EBI5- 0005-03), and Federal Ministry of Education and Research (Germany, grant no. 16LC2021A). The funders had no role in study design, data collection and analysis, decision to publish, or preparation of the manuscript.

**Competing interests:** The authors have declared that no competing interests exist.

range from 1 to 4 million m$^3$ of wood per year [8, 9]. While climate warming may lead to reduced levels of snow disturbances [10] the future changes are likely to be spatially asymmetric. For example, snow damage is projected to decrease in southern and western Finland but in northern and eastern parts of the country heavy snow loads are expected to increase. This is because the warmer and more humid climate will increase the occurrence of wet snow hazard events and conditions favorable for rime accumulation in these areas [11, 12].

Snow disturbances are an inherent part of the forest ecosystem in northern and high-altitude forests. They cause economic losses in terms of damaged wood and increased tree mortality [8]. Snow disturbances in forests also damage the infrastructure; the power grid in particular is vulnerable as tree tops and trees with heavy crown snow loads fall on the power lines [13]. Snow damaged trees and areas are also more susceptible to subsequent damage by insects or fungi [8]. Many of the negative effects of snow disturbances could potentially be alleviated by improved planning and forest management, but this requires accurate information about the damage risks. Spatial risk information is increasingly required by the society and it is used actively in management operations and financial planning among owners, industry, and insurers.

Precise forest and climate data has made it possible to present risk information at high resolution. For example, Suvanto et al. [14], mapped forest wind damage probabilities in forests at 16 m x 16 m resolution, by using a model that drew from damage observations made in the Finnish national forest inventory (NFI), spatially identified high wind areas, and environmental and forest resource data from various open data sources. The high spatial resolution of the map allows the consideration of disturbance probability on the level of individual forest stands, i.e. the spatial unit in which the management decisions are being made. In the disturbance dynamics of northern forest ecosystems, snow disturbances play an important role, and therefore a better understanding of how snow damage risks can be predicted at large scale but at high resolution is needed.

Snow damage to trees is induced when the forces generated by a large crown snow load, often together with wind, exceed the force required to break the stem of the tree. Meteorological data is crucial in modelling forest snow disturbances, as specific meteorological conditions are needed for snow to accumulate on trees The accumulation occurs typically within a narrow temperature range close to 0˚C [15]. Conditions after the snowfall are important for the damage, as retention of snow in the tree crowns is temperature dependent [8]. As the accumulation of rime and snow on trees is driven by temperature and wind conditions, topographic factors are correlated to the occurrence of snow damage [8, 16]. Snow load on trees can be categorized in different types, such as rime, wet snow, dry snow and frozen snow, and the physical process of snow accumulation differs by the type. Lehtonen et al. [16] showed that improved results in modelling snow load in tree crowns can be achieved by considering the different snow load types separately.

The characteristics of the forest stand and the trees play an important role, as damage occurs when the gravitational forces and torque caused by the crown snow load exceed the stem tolerance limit. The tolerance is largely related to stem taper and characteristics of the tree crown, while these are driven by factors such as tree species and stand characteristics [8, 17]. From a biomechanical perspective, older trees with stronger stem taper and thicker stems should be more resistant to crown snow loads than smaller trees with modest stem taper and thinner stems. The density of stand may indirectly affect the susceptibility of trees to damage, as density-driven competition drives the growth of thin and tall stems [8, 17].

Coniferous species are generally more susceptible to snow damage than deciduous trees, and Norway spruce is less vulnerable compared to Scots pine [8, 18]. Tree structural properties predisposing trees to damage vary also within species. In Norway spruce, the tree morphology

varies across the species range so that in high altitude and latitude areas the narrow crown shape and dense, horizontal branches reduce the accumulation of snow on the crowns, decreasing the probability of snow damage [19–21].

In this study, our aim was to (1) assess the importance of meteorological and topographic factors versus forest properties for the occurrence probability of snow damage in forests, comparing results from winters with typical snow load conditions and an exceptionally heavy snow load winter of 2017–2018 [22], and (2) produce a snow damage probability map for Finland and test the ability of the map to identify the stands vulnerable to snow disturbances. As the meteorological variable, we used model-derived crown snow load, which should be the best proxy for damage-causing climatic conditions and which allows predicting changes of snow damage risks with the changing climate.

## Materials and methods

### National forest inventory data

National forest inventory (NFI) data was used for the snow damage observations and for the forest characteristics data. The used data included plots from the 10th (2005–2008), 11th (2009–2013) and 12th (2014–2018) Finnish NFIs [23, 24]. NFI10 measurements from 2004 were excluded as no full 5 year period of snow load data was available before that year. To avoid having repeated measurements from the same plots in the data, only temporary NFI plots from NFI10 and NFI11 were included in the analysis, whereas all plots (temporary and permanent) were included from the NFI12. Only NFI plots on forest land were included and plots on treeless stands were excluded from the data. Data points with missing data in any of the used predictor variables were excluded in the analysis. The final data consisted of a total 111 677 plots, in 2 380 of which snow damage was recorded (Table 1).

**Table 1. Statistics of stand level snow damage, damage severity and damage type in the NFI data.** The values are also shown separately for 2005–2017 with typical snow conditions and year 2018 with exceptionally high snow loads.

|  | All | 2005–2017 | 2018 |
|---|---|---|---|
| Total number of plots | 111 677 | 102 671 | 9 006 |
| Total damaged plots | 2 380 | 1 885 | 495 |
| % damaged plots | 2.13 | 1.84 | 5.50 |
| Damage severity (% of cases)* |  |  |  |
| 0, slight damage | 57.4 | 59.4 | 49.9 |
| 1, moderate damage | 38.9 | 37.3 | 44.8 |
| 2, severe damage | 3.7 | 3.2 | 5.3 |
| Damage type (% of cases) |  |  |  |
| Dead standing trees | 0.5 | 0.5 | 0.6 |
| Uprooted or broken trees | 75.8 | 75.6 | 76.4 |
| Other stem damage | 0.5 | 0.5 | 0.4 |
| Dead or broken crowns | 12.3 | 10.6 | 18.8 |
| Other crown damage | 10.7 | 12.5 | 3.6 |
| Branch damage | 0.2 | 0.2 | 0.2 |
| Defoliation | <0.1 | 0.1 | – |
| Discolouration | <0.1 | 0.1 | – |

* slight damage–does not affect silvicultural quality or change the development class of the stand, moderate–lowers the quality of the stand by one class (e.g. from good to satisfactory), severe–decreases the quality of the stand by more than one class, changes the development class to unstocked or makes low-yielding stand significantly less productive.

Stand level snow damage observations from the NFI data were used in the study. We included in the analysis all damage cases that occurred in the dominant tree storey of the stand (i.e., the tree storey that determines silvicultural operations for the stand), where the causal agent of the primary damage had been classified as "snow" and the timing of the damage was estimated to be within 5 years. The definition of the damage agent and the estimation of time since damage are based on the judgement of the NFI field team on site.

The damage type was most often fallen or broken trees (no distinction of these two are made in the data) but also other damage types were found (Table 1). Damage severity is recorded in the NFI as a cumulative effect of all damage agents found in the stand, and no information about the severity of snow damage specifically is included if also other damage agents are present. Severity is assessed on a four-point scale (0 to 3) and most stands with snow damage are classified to the two lowest classes (0 = slight damage, does not affect the silvicultural quality of the stand or change the development class, and 1 = moderate damage, lowers the silvicultural quality of the stand by one class), with some observations in the second highest class (2 = severe damage, decreases the quality of the stand by more than one class) and no observations in the highest damage severity class (3 = complete damage, immediate regeneration required). See [25] for detailed description for documentation of damage in the Finnish NFI. Table 1 shows the distribution of the damage observations in the different damage type and severity classes. In our analysis we grouped all the snow damage observations together and therefore the analysis does not consider differences in damage type or severity.

Other information from the NFI used in our analysis included average tree height and diameter at breast height (DBH) in stand, basal area, previous forest management operations and their timing, site type and proportions of basal area represented by different species (Table 2). The exact variables derived are described in the "Statistical modelling" section.

Stand average DBH was not recorded for stands of development class "young seedling stand", where the height of the dominant tree species is less than 1.3 meters. For these stands the stand average DBH was set to 0 cm, as the seedling do not reach the breast heigh (1.3 m). In NFI10, DBH was also missing for the development class "advanced seedling stand". For these, the DBH was estimated based on the measurements in NFI11 and NF12. DBH in this

**Table 2. Number of plots and the descriptive statistics for forest, topographical and snow load variables included in the final model for damaged and non-damaged plots separately and for all the plots in the data.** Values for categorical variables show percentages of plots in each class. Values for continuous variables show mean and standard deviation, the latter in parenthesis.

| | Description | Damaged | Non-damaged | All |
|---|---|---|---|---|
| Number of plots | | 2 380 | 109 297 | 111 677 |
| FOREST | | | | |
| Species | dominant species of the stand | | | |
| *pine* | | 73.1% | 61.4% | 61.6% |
| *spruce* | | 19.3% | 27.7% | 27.6% |
| *other* | | 7.6% | 10.9% | 10.8% |
| DBH (cm) | stand average DBH | 16.1 (5.8) | 16.1 (8.6) | 16.11 (8.57) |
| BasalArea ($m^2$ $ha^{-1}$) | basal area of trees | 18.9 (8.0) | 16.8 (9.6) | 16.8 (9.5) |
| NorthBoreal | Plot located in the north boreal zone | 20.1% | 12.3% | 12.5% |
| ABIOTIC | | | | |
| Snowload (kg $m^{-2}$) | max crown snow load, within 5 years before the NFI measurement | 64.6 (29.6) | 49.3 (16.2) | 49.7 (16.7) |
| SnowloadLongterm (kg $m^{-2}$) | Average of winter maximum snow load in 2000 to 2015 | 38.1 (6.8) | 34.5 (7.40) | 34.6 (7.4) |
| RelativeElevation (m) | difference to mean elevation in 1 km radius | 3.1 (9.5) | 1.2 (7.3) | 1.2 (7.4) |
| Altitude (m.a.s.l.) | altitude from sea level | 165.3(73.1) | 130.5 (68.7) | 131.2 (69.0) |

development class was predicted based on average tree height and dominant tree species by fitting a GLM model with gamma distribution and log-link function to the NFI11 and NFI12 data where the DBH was available, and then using this model to predict the DBH values for the advanced seedling stands in NFI10 where the DBH information was missing.

### Snow load on tree crowns

Maximum snow load on tree crowns was calculated for each winter for years 2001 to 2018, using the snow load model of the Finnish Meteorological Institute (FMI) [16] and the ERA5 reanalysis data [26].

The snow load model is a statistical model in operational use at the FMI. The model assumes a tree with cone-shaped crown with a projected catchment area of one square meter from above and from the side in the direction of the wind and calculates the snow load on tree canopies in four different snow accumulation types: rime, dry snow, wet snow and frozen snow [16]. Here, the sum of the different snow load types was used, and the maximum snow load of the previous five years from the NFI measurement date was used for each NFI plot, as the snow damage observed on the plots may have occurred within 5 previous years.

### Topographic variables

Altitude as meters above sea level was extracted for the NFI plot locations from the 25 meter resolution digital elevation model (DEM) from the National Survey of Finland. Relative elevation was calculated from the same DEM as the difference between the altitude at the plot location and the average altitude within one kilometer radius. Thus, negative values of the variable represent topographic positions lower than the near surroundings and positive values higher.

### Statistical modelling

Statistical models were fitted using the occurrence of snow damage in the NFI plots as the binary response variable and forest properties, snow load data and topographic variables as predictors. Only snow damage cases that had occurred within 5 years of the NFI field measurement date (according to the estimate of the field team) were considered.

Two different types of statistical modelling methods were used: generalized linear models (GLM) and generalized additive models (GAM), both with a logistic link function. GAM is an extension of a GLM where the linear predictor contains a sum of smooth functions of continuous predictors. Using smooth functions instead of detailed parametric relationships (as done in GLM) allows for more flexibility in the form of the dependence of the response variable on the predictors [27].

The potential predictor variables considered in the model selection were chosen based on the existing understanding of factors affecting snow damage probability in forests [8, 15–19]. The variabels were grouped into abiotic variables relating to snow load and topography (ABIOTIC) and forest variables (FOREST). The ABIOTIC variable group contained variables describing crown snow load (maximum of previous 5 years before NFI field measurement), long term average of winter maximum crown snow load (2000–2015), altitude from sea level, relative elevation in comparison to mean elevation within a kilometer radius and a variable describing if the plot was located in the north boreal vegetation zone, according to the biogeographical zones data from the Finnish Environment Institute [28].

In the FOREST variable group, the potential predictors considered in the model selection included dominant tree species of the stand, average DBH, average tree height, stand basal area, forest management history, site type (poor vs fertile, using the same classification as in [14]), number of tree species, proportion of basal area by the most abundant species and the

Shannon diversity index, calculated from the proportions of basal area by each species. For forest management history, three different variables were included—all thinnings, pre-commercial thinnings and tending of seedling stands. All were included as presence/absence variables that described if the management operation had been carried out at the stand more than 5 years ago. Management information within five years from the NFI measurement was not considered because, if snow damage had occurred in the stand, it would not be clear if the management was done before or after damage (damage was considered from the latest 5 years). To find potential species-specific responses, interaction terms were tested between tree species and DBH, basal area, the snow load variables and the north boreal zone variable.

The model selection was done using only the GLM model. The model predictors were chosen from the pool of potential predictors described above based on (1) the existing understanding of how abiotic factors and forest properties affect snow damage probability in forests, (2) availability of national extent GIS-data to be used for map prediction, (3) statistical significance of highest order terms in the model, requiring significance on the level of $p < 0.01$, as the large sample size easily leads to small p-values, (4) improvement in AIC when comparing alternative models and (5) collinearity between predictors, determined by the generalized variation inflation factor (GVIF). If the GVIF exceeded 4 for any of the predictor variables, one of the correlated variables was left out of the model. The decision on which variable to exclude was made following the same five steps of comparing alternative models. For continuous variables with non-negative values, log-transformations with natural logarithm were tested and included where they led to a lower AIC. For transparency of the model selection process, intermediate model versions with variables not included in the final model can be found in the (S1 File). While GVIF was used to assess multicollinearity of predictors in the model selection process, correlation matrix of the continuous predictors in the final model is found in Fig 1 to support the interpretation of the model results.

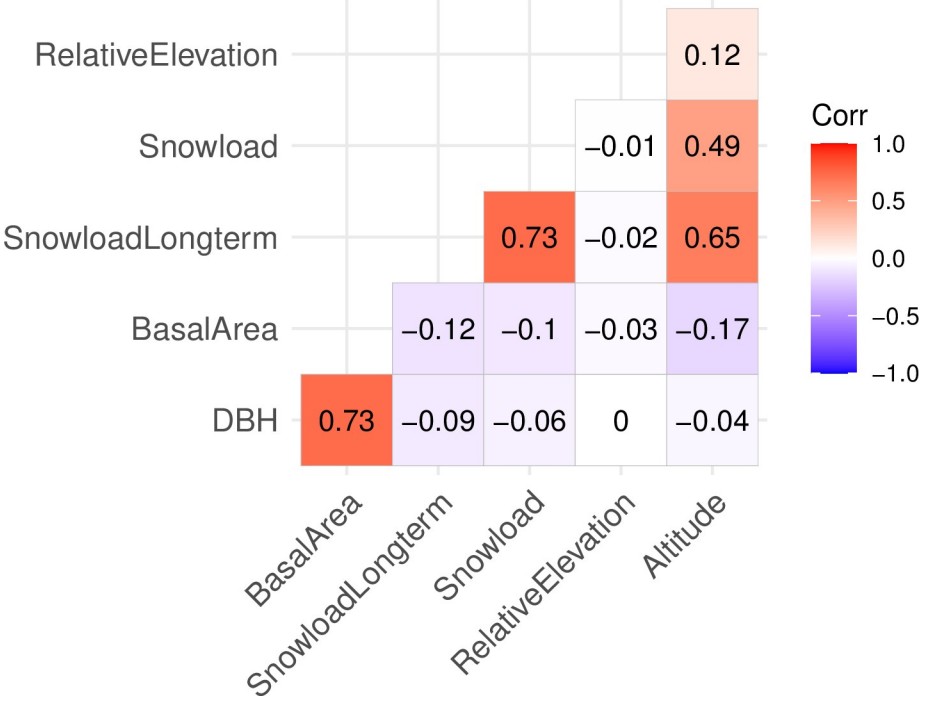

**Fig 1. Correlation matrix of the continuous predictors in the final model.**

After the predictors were selected for the model, two additional submodels were formed to have three models: a full model with all selected predictors (FULL), and two models with only a subset of the predictors in the full model: a model with only abiotic predictors (ABIOTIC) and a model with only predictors related to forest properties (FOREST) (see variables included in each group in the final model in Table 3, results for the variables not included in the final model can be found in S1 File). In case of an interaction between variables in different variables groups, both variables were included in the FOREST group.

Models with the same predictor variables were then fitted as generalized additive models (GAM) to test if using a non-parametric model would lead to better outcome, as they are able to effectively deal with non-linear relationships. Continuous predictor variables were included in the GAM models as smoothing spline functions. The dimension parameter (k), that sets the upper limit on the degrees of freedom related to the smooth, was set to 15 for all variables. The suitability of the k parameter was assessed visually. In addition, the effective degrees of freedom after fitting the model were lower than k for all of the terms, suggesting that the chosen k values were sufficiently large.

The performance of the models was assessed with 10-fold stratified cross-validation, where the number of damaged plots was divided evenly into the folds. One fold at the time was used as test data while the model was fitted with the remaining nine folds. Receiver operating characteristic (ROC) and area under curve (AUC) were calculated for the test data to assess the model performance. AUC value of 0.5 corresponds to a situation where the model does not do better than randomly assigning the prediction values, whereas AUC value of 1 would mean that the model is perfectly able to discriminate between damage cases and no-damage cases. As a rule of thumb, 0.7 is often used as an acceptable level of discrimination between the classes [29].

To compare the results for typical snow load winters and an exceptionally high snow load winter, AUC values for the cross-validation were calculated in three different subsets: (1) using all the data in the test data fold, (2) using only data from 2005–2017 in the test data fold ("typical snow load winters") and (3) using only data from the 2017–2018 winter ("exceptional snow load winter", Fig 2). Models were always fit with data from all years and the above described subsets were only used in the test folds.

**Table 3. Model results for the full GLM model.** See full descriptions and units of the covariates in Table 2.

| | Estimate | Std. Error | z value | Pr(>\|z\|) |
|---|---|---|---|---|
| Intercept | -7.209 | 0.168 | -42.975 | < 0.001 |
| SpeciesSpruce[1] | -0.287 | 0.058 | -4.952 | < 0.001 |
| SpeciesOther[1] | 0.716 | 0.214 | 3.350 | < 0.001 |
| DBH | -0.072 | 0.004 | -16.308 | < 0.001 |
| log(Basalarea + 0.5) | 1.101 | 0.048 | 22.721 | < 0.001 |
| NorthBoreal | -0.031 | 0.069 | -0.450 | 0.65 |
| SnowloadLongterm | -0.027 | 0.005 | -6.093 | < 0.001 |
| Snowload | 0.032 | 0.001 | 30.822 | < 0.001 |
| RelativeElevation | 0.026 | 0.002 | 10.752 | < 0.001 |
| Altitude | 0.006 | 4.7E-04 | 11.892 | < 0.001 |
| SpeciesOther x DBH | -0.092 | 0.016 | -5.670 | < 0.001 |
| SpeciesSpruce x NorthBoreal | -0.749 | 0.186 | -4.029 | < 0.001 |

[1] Compared to the reference species Scots pine.

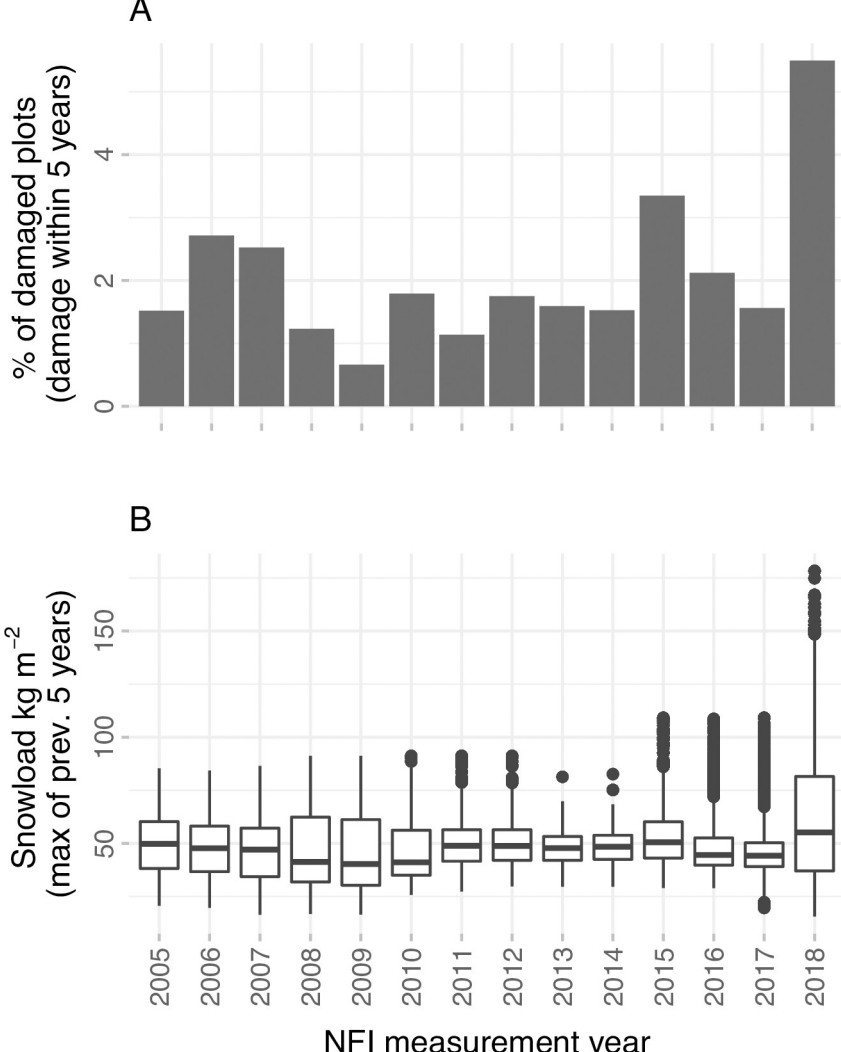

**Fig 2.** Percentage of plots with snow damage in each year (A). Year refers to the year the NFI plot has been measured on the field, damage may have occurred within previous five years). (B) Maximum snow load at the NFI plots within a five year time window.

Statistical modelling was done in R version 3.5.2. [30], ROC and AUC were calculated with the R package *pROC* [31], the GAMs were fitted using the R package *mgcv* [27].

## Mapping of damage probability

The snow damage probability map was calculated for the whole country of Finland in 16 x 16 m pixel resolution, by using the full GLM and GAM models and geographic information system (GIS) datasets representing the predictors of models.

Regarding GIS datasets, multi-source forest inventory (MS-NFI) forest resource maps for 2017 [32] were used for the forest variables (tree species, DBH, basal area). Topographic variables (altitude and relative elevation) were derived from the 25 meter resolution DEM of the National Land Survey of Finland and resampled using bilinear interpolation to the same 16 m x 16 m grid. Snow load data [16] for winter 2017–2018 was used in the calculation of the map, as this winter was also used for the testing of the map.

The processing of GIS data was conducted using R (package raster), Python and GDAL. The calculation of the map was done using R packages raster [33] and sp [34].

## Testing the map

The test data for the damage risk maps for winter 2017–2018 was obtained from the Finnish Forest Centre.

For damage events, forest use declarations where snow damage had been recorded were extracted from the data, using the reports sent to the Forest Centre from December 1st 2017 to September 30th 2018. Forest owners are required by law to submit a forest use declaration to the Forest Centre before conducting forest management operations at their stands and since 2012 these declarations have included information about forest damage in the stand in case the damage has been the reason for the logging operation. The declarations contain information about the stand, including the type of the occurred damage with a separate class for snow damage, and a spatially referenced polygon outlining the stand. The final test data contained a total of 11 807 snow damaged stands (referred to as "snow damage polygons" from now on).

To compare the snow damage polygons from forest use declarations to non-damaged stands, we used another data set by the Forest Centre, which contains spatial polygons and basic forest property information for forests on private lands in Finland. This database is based on a combination of data from different sources, such as remote sensing, field measurement and reports from forest owners [35]. From this data, one percent of the polygons in the whole country was randomly sampled. Polygons classified as open stands (i.e., did not have trees) were excluded from the sample. While this data set does not contain information about forest damage, we assume that these stands are not damaged. The resulting data consisted of 101 073 polygons (referred as "non-damaged polygons" from now on).

To test if the map was able to differentiate between damaged and non-damaged stands within the larger damage area (as compared to only differentiating the general damage area from the rest of the country), another test was carried out by only including the non-damaged polygons that were located within 10 kilometers from the damaged stands. This subset contained 16 486 non-damaged polygons.

For both snow damage polygons and non-damaged polygons the average value of snow damage map pixels within each polygon was calculated for both maps based on GLM and GAM models. Then, the distribution of the map values was examined on the snow damaged and non-damaged maps, and ROC curves and AUC values were calculated to assess the performance of the maps to identify the snow damage cases.

Both of the used data sets (forest use declarations and stand polygons for private lands) are published by the Finnish Forest Centre under CC BY 4.0 licence and are openly available (https://www.metsaan.fi/paikkatietoaineistot). The used data were downloaded in October 2020.

## Results

The GLM model results show that abiotic factors, especially crown snow load, drive the snow damage, as damage probability increases with increasing snow load, relative elevation, and altitude (Table 3, Fig 3). Yet, forest characteristics also have an impact on damage occurrence. Damage probability was higher in stands with higher basal area and in stands with lower average DBH. The model showed higher damage probabilities in stands dominated by pine compared to other species. Norway spruce dominated stands show regional different patterns, with disturbance probability being significantly lower in the north boreal zone compared to other

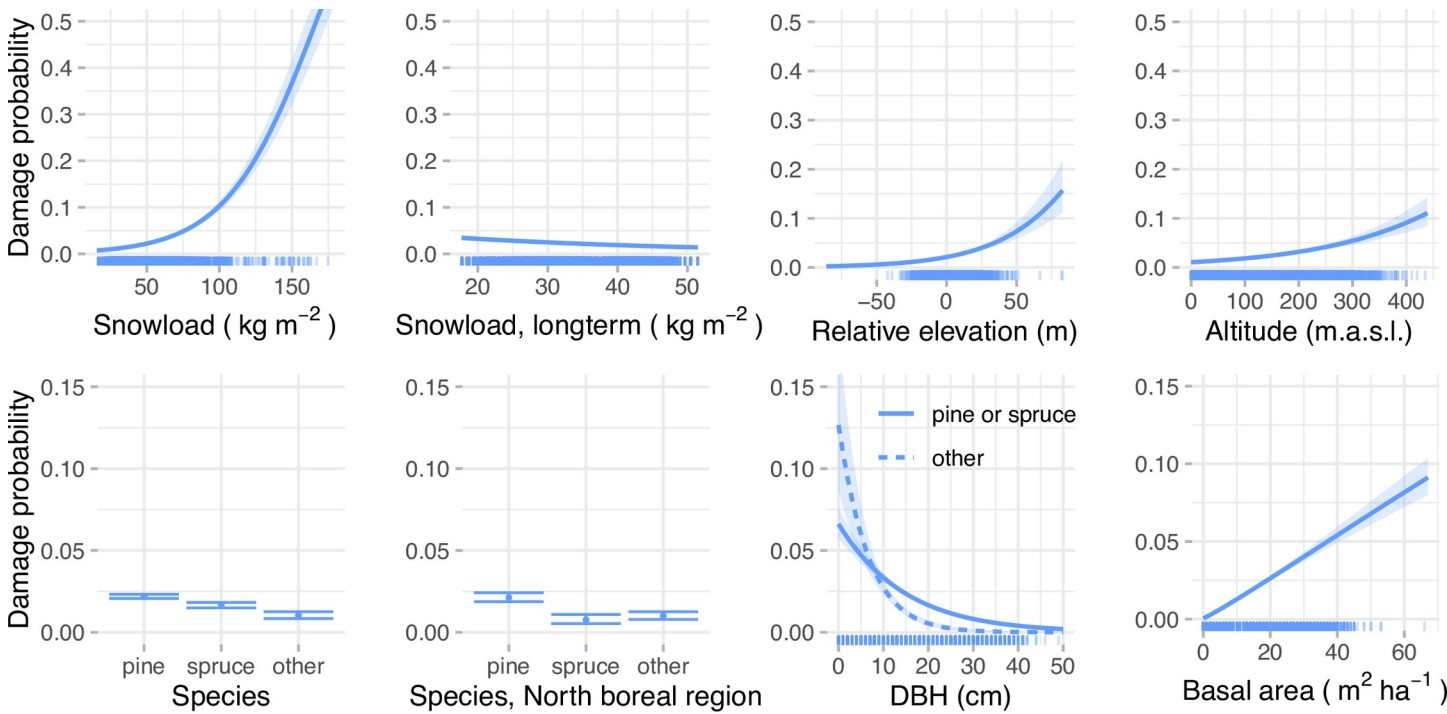

**Fig 3. The impact of predictors for the probability of snow damage occurrence according to the full GLM model.** Note different y-axis limits in abiotic variables (upper row) and the forest variables (lower row). The rug showing the distribution of data points is a random subset of 10 000 plots from the original data.

parts of the country. For species group "other", mainly consisting of birches, higher values of damage probability were predicted for small DBH stands compared to pine and spruce (Fig 3).

The GAM models showed generally similar patterns as GLM models but also revealed nonlinearities not visible in the GLM results. For example, probability of damage only started to rise drastically with snow load after 75 kg m$^{-2}$ (Fig 4), which is clearly higher than the snow loads observed in typical winter conditions (Fig 2B). The GAM results also show decrease of damage probability with relative elevation and altitude after certain thresholds, but as there are few observations at high values of both of these variables, there is high uncertainty of the shape of the spline. Long term snow load (15 years average) also showed a nonlinear trend with the damage probability, with damage probability values peaking at 30 kg m$^{-2}$.

In the model selection process variables were left out of the model based on three reasons: (1) high p-values (p > 0.01) or no improvement in AIC, (2) high variance inflation factors, suggesting high multicollinearity between variables and (3) a lack of national level wall-to-wall spatial data available for the creation of the snow damage probability map. Potential predictors excluded from the final model based on p-values and AIC values were variables describing thinning and precommercial thinning in the stand, site type, number of species and percentage of basal area covered by the dominant species. High GVIF affected the model selection when both DBH and average tree height were included in the model. To decide which one of these two would be included in the final model, we compared AIC values of models with DBH but no height and with height but no DBH. The model with DBH showed lower AIC value and therefore DBH was selected for the final model instead of tree height. Lack of available spatial data led to exclusion of two variables from the model that could have been included based on the other model selection criteria. These were variables describing tending of seedling stands (negative coefficient estimate, i.e., lowering the damage probability, p = 0.002) and Shannon diversity index of tree species, calculated from the species-specific share of basal area (negative

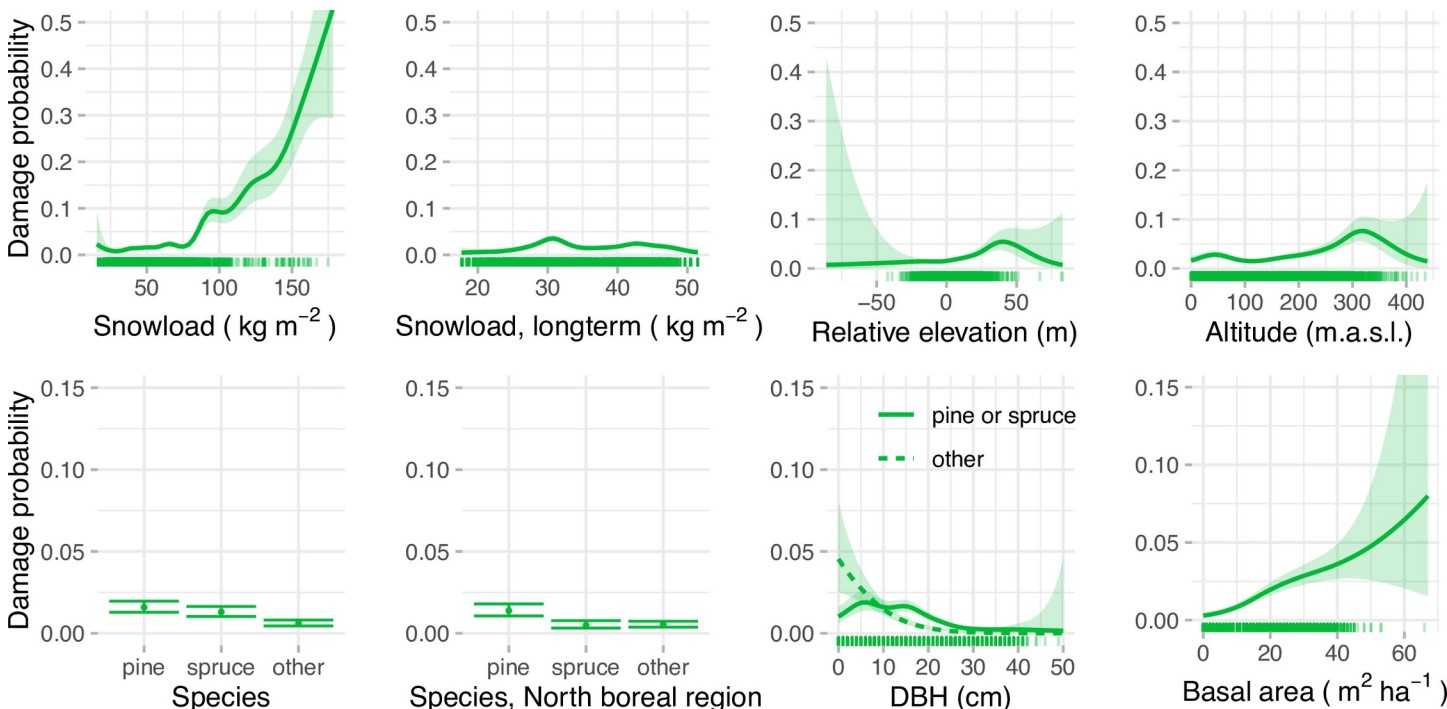

**Fig 4. Effects plots for predictors in the full GAM model.** Note different y-axis limits in abiotic variables (upper row) and the forest variables (lower row). The rug showing the distribution of data points is a random subset of 10 000 plots from the original data.

coefficient estimate, i.e. lowering the damage probability, p = 0.004). Full results for models with variables not included in the final model are found in (S1 File). The NorthBoreal variable is included in the model despite the high p-value because the higher order term, the interaction term SpeciesSpruce x NorthBoreal, has a p-value < 0.001 (Table 3).

The cross-validation of the models showed that the FULL model with both abiotic and forest variables included performed better than the submodels with variables from only one group included (models ABIOTIC and FOREST, Fig 5). There was a difference between cross-validation results of winters with typical snow load conditions (2005–2017) and the 2017–2018 winter with exceptionally high snow loads. In the 2017–2018 winter the AUC values were also notably higher than in the results with full data or only years 2005–2017 and the ABIOTIC model with only abiotic predictors performed nearly as well as the full model (Fig 5). In the cross-validation, the GLM and GAM models gave rather similar results. In general, GAM performed better for the ABIOTIC model and GLM for the FOREST model (Fig 5).

The snow damage probability maps predicted the highest snow damage risks in 2017–2018 near eastern border of the country (Fig 6). The overall patterns in GLM and GAM maps were similar, with only minor differences. Testing the map with snow damage polygons showed that the model is able to predict damage probability on acceptable level also when spatial data sets with national wall-to-wall coverage are used for prediction instead of the field-measured NFI data originally used in model fitting (Fig 7). Very high AUC values were obtained when the non-damage polygons were randomly sampled from the whole country (Fig 7A) but also the test with non-damaged polygons sampled only from proximity of damaged polygons showed good ability of the model to identify the snow damaged polygons (Fig 7B). The test showed quite similar results for the two modelling methods, though the map produced with the GAM model gained slightly better results (Fig 7).

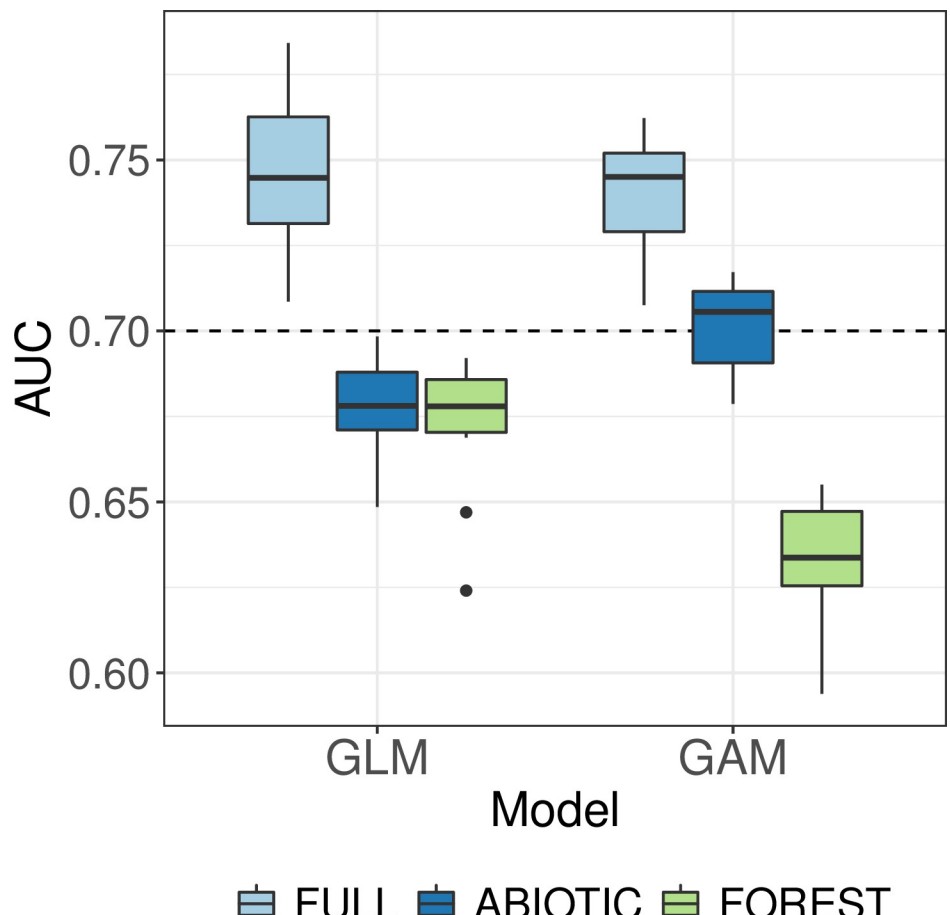

**Fig 5. Cross-validation results for GLM and GAM with different predictor sets and for different time periods.**
Dash line shows the AUC = 0.7 threshold for acceptable level of discrimination between cases and non-cases.

## Discussion

We quantified the role of critical meteorological conditions to the snow damage risks by combining estimates of crown snow loads to the actual measurements of forest properties and snow damage from a large area in the boreal zone. The results showed that snow load becomes the dominating driver of damage during heavy snow years, but forest properties still improve the prediction of damage. During regular winters with typical snowpacks, the importance of forest properties in identifying risk locations is emphasized. Further, we demonstrated that the damage locations can be reliably pinpointed, especially on heavy snow years, at high resolution, which can be used to facilitate salvage logging and conservation planning. Moreover, the snow damage risk model can be applied with data of long-term snow load return-rates or projections of future snow loads, to generate risk estimates for the forest development scenarios under climate change.

The best predictions of snow damage probability were obtained when both abiotic variables (long term and recent snow load and topographic variables) and forest characteristics (species including an interaction with location in north boreal zone, DBH, basal area) were included in the model. By combining forest related predictors with snow load information from the winters preceding the NFI observations, our work extends further from many previous snow damage studies focusing solely on forest and site characteristics [2, 5, 36]. While studies focusing

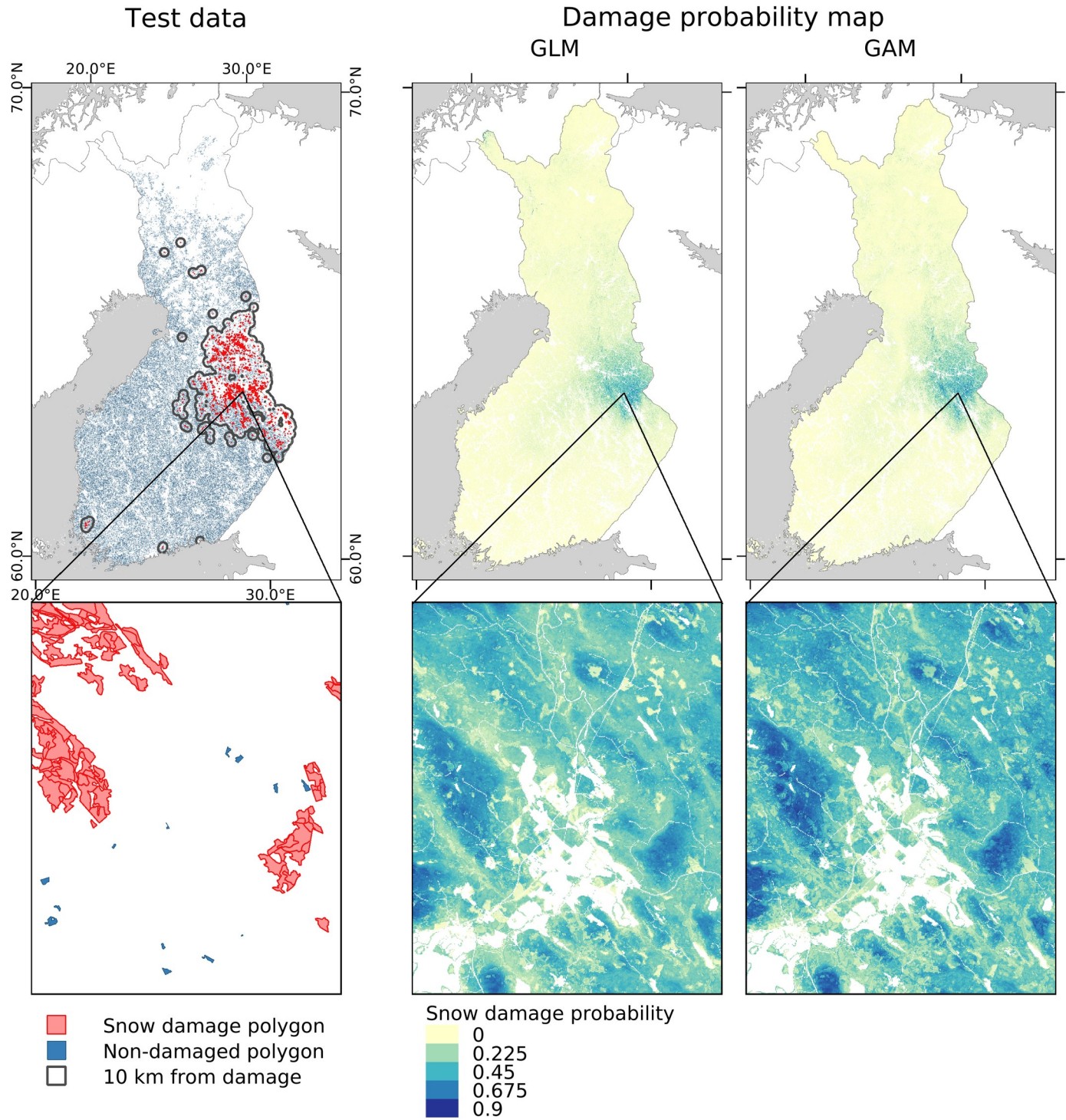

**Fig 6. The Forest Centre data (published under CC BY 4.0 licence) used for testing the snow damage probability maps, and the snow damage probability maps calculated with the snow load data from winter 2017–2018, using the full GLM and GAM models.** Background map: Natural Earth (public domain).

on single snow damage events have been able to include both forest and snow information before [6], this is not the case for studies using long-term data from several damage events. In addition, the data describing snow load in the tree crown [16], used in our analysis, provides a

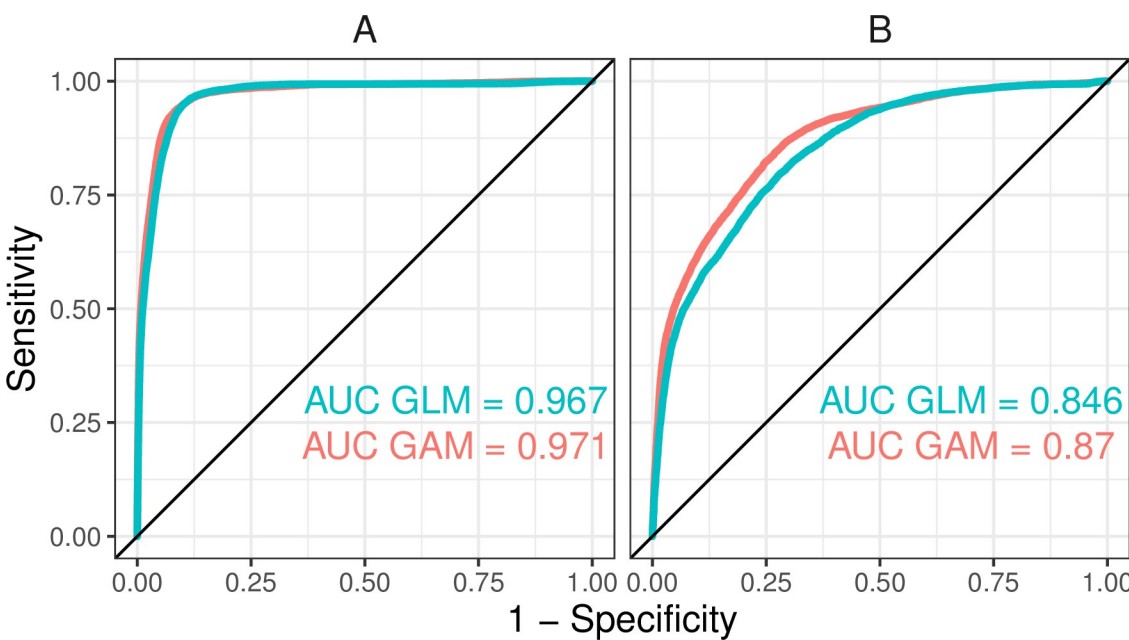

**Fig 7.** ROC curves and AUC values for the test of the snow damage probability map with the Forest Centre data: using non-damaged polygons from the whole country (A) and only considering non-damaged polygons within 10 km distance from snow damaged polygons (B).

more realistic presentation of damage conditions compared to using information about snow depth, as used by, for example, Hlásny et al. [6]. On the other hand, our work offers improvements compared to using only the meteorological estimation of snow load that assumes a theoretical constant shape of tree crown [11, 16], by combining it with detailed information about forest properties. This opens new practical application possibilities, as increased accuracy in snow damage probability calculations can be attained when combining high-resolution forest data to the estimates formerly based only on simplified assumptions of the tree properties.

The exceptionally heavy snow load winter 2017–2018 showed distinctively different patterns in our results compared to winters with lower snow load levels, as the model performed clearly better for the heavy snow load winter and the abiotic variables alone contributed for most of the model performance. This suggests that the processes of snow damage between heavy snow load winters and typical winter conditions have dissimilarities. It seems that during winters with low or moderate snow loads, snow disturbances only occur in the most vulnerable forests, emphasizing the importance of the forest predictors in these conditions. On the other hand, in winters with exceptionally high snow loads, damage can occur also on forests not as sensitive to snow damage, which is reflected in our results by the increased importance of the abiotic predictors. With lower snow loads forest properties drive the snow damage probability while their relative role diminishes when snow loads rise to exceptionally high levels. This interpretation is further supported by the effect of snow load in our results starting to strongly increase only after approx. 75 kg m$^{-2}$, a level of crown snow load only rarely occurring in typical winter conditions (Fig 2B). Uncertainties related to snow load data as well as the NFI damage observations may, at least partially, also play a role in the difference between typical and extreme snow load years. During a heavy snow load winter, the snow damage in forest is likely to be clearer and less likely by the NFI field team to be mistaken for wind damage. Meteorological estimation of snow loads may also be less uncertain when the snow loads are high.

The effects of abiotic and forest factors on snow damage probability in our final model were largely in line with the previous research. Increasing damage probability with elevation from sea level and with relative elevation from the surrounding terrain are backed by previous results [8, 16, 37]https://www.zotero.org/google-docs/?pVHvTG. In accordance with literature a review by Nykänen et al. [8], damage probability in our results increased with basal area and decreased with stand average DBH. In our results, the effect of DBH was different for stands not dominated by pine or spruce (i.e., mainly birch dominated). Our results supported earlier research showing higher susceptibility to damage in coniferous versus deciduous trees and in Scots pine compared to Norway spruce [2, 8, 38].

In the interpretation of model results it is important to consider the correlations between the model predictors. For example, if considering forest stands with the same DBH, the stands with higher basal area have a higher snow damage probability (as illustrated in Figs 3 and 4). However, these two variables are positively correlated, meaning that an increase of basal area is usually accompanied with an increase in DBH, which in turn has a negative effect on the damage probability. Therefore, the higher basal area leads to increased damage probability only if the change in DBH is not large enough to counter this effect. Similarly, snow load, long term snow load and altitude are correlated, and their effects should be considered together when interpreting the model coefficient estimates.

Our results reveal patterns suggesting adaptation of forests to high snow loads. First, in addition to the differences in damage probability between species, our results revealed geographical differences within Norway spruce, with spruce stands in the north boreal zone showing reduced probability of damage compared to spruce stands in the other parts of the country. The spruce trees in high latitude and altitude areas are known to have different crown morphology, with narrow crown shape reducing the accumulation of snow load on trees [20, 21]. Our results show in practise how the morphological variation in the species leads to geographical variation in the predisposition of the trees to snow damage. Second, the negative effect of long-term snow load on the damage probability in the model suggests that forests in areas with historically higher snow loads are more resistant against snow damage. This effect was not species-specific but instead seems to affect stands regardless of the dominant species, as the interaction between long-term snow load and species was not statistically significant (S1 File). While the morphological differences may play a role also here, differences in forest structure in areas with high snow loads may also contribute to explaining this result, if the basal area and DBH included in the model are not sufficiently accounting for stand structure.

We did not find a statistically significant connection between thinnings and damage probability (results in S1 File). This finding contradicts previous research. According to literature review by Nykänen et al. [8] trees in unthinned stands are more susceptible to snow damage and delayed thinning increases the snow damage risk. On the other hand, thinnings are found to temporarily increase the susceptibility of trees to snow damage, leading to higher damage risks during the first and second years after thinning [8, 38] and Wallentin and Nilsson [39] found snow damage to be positively correlated with thinning intensity. We would not conclude from our results that forest management does not affect snow damage probability, instead the non-significance of the management effect is likely to be related to the insufficient detail of the used data. The main weakness in our analysis in regard to forest management is the imprecise definition of time between the damage and the management operation. As damage in our analysis had occurred within a time-window of five years before the NFI measurement, management operations could only be included if they had occurred more than five years ago. Otherwise, it would not have been possible to differentiate between thinnings before the damage from ones occurring only after the damage. However, with this approach we are likely to lose the most sensitive period of one to two years after the thinning, when the damage

sensitivity is the highest [8, 38]. It is also worth noting that the forest variables included in the model (DBH and basal area) are strongly affected by management and, therefore, the effects of management are not completely excluded from the model.

Many previous studies have analyzed snow and wind damage together [36, 40–42] as these two processes can act jointly in a damage event. For example, wind can more easily break trees with heavy snow load, or strong winds can either increase the snow accumulation or prevent the accumulation of snow on trees by shedding the snow from the branches [8, 15]. However, while these processes can be related to each other, our results here and the previous results for wind damage [14] show that snow damage and wind damage affect different types of forest stands and have also spatially different occurrence patterns. While wind damage risk increases with tree height [14], snow damage is more typical in smaller trees, as shown in our results. In addition, snow damage can also occur with a minimal effect of wind (as in [6]) and wind disturbances often are not accompanied by snowfall.

The challenge in considering wind and snow separately in NFI data is in reliably identifying the cause of the damage in the field when field measurements are not targeting any specific damage event and stem breakages and uprooting can be related to either of the damage causes or their combined effects [36]. For example, in southern Finland where heavy snow events are less common, snow damage may be mistakenly classified as wind damage, as those are more common in the region. The damage may have occurred already several years before the field measurement, making the correct identification of damage cause even harder. This adds uncertainty in the analysis and may also partly contribute to our results on why the model did not perform as well for the winters without heavy snow loads. Yet, while this uncertainty needs to be acknowledged we argue that, due to the differences in the two disturbance processes, it is beneficial to study damage caused by wind and by snow separately, whenever the used data makes this possible.

In our analysis, we did not differentiate between different snow damage types and, even though the used NFI data did contain some information on the damage type (see Table 1), stem breakage and uprooting were pooled in the same class, thus preventing us from analyzing them separately. This is a potential shortcoming of our analysis, as different damage dynamics may be behind stem breakage versus uprooting [6, 17, 42].

Logistic regression models (GLM) have long been the traditional method for modelling snow and other forest disturbances [36, 40–42] whereas GAM provides more flexibility in modelling non-linear responses, as the relationship between continuous predictors and the response variable can be modelled with smoothing spline functions instead of the linear relationships [27]. In our results, the comparison of the two statistical modelling methods showed rather similar results for the full model despite the method used. The GAM model performed better for the ABIOTIC model and the GLM for the FOREST model. This difference is likely to explain the better performance of the map based on the GAM model for the test data from winter 2017–2018, since this was a high snow load winter where the abiotic factors drove the damage probability. While the flexible smoothing spline functions in GAM increased the model performance in the case of the abiotic predictors, the traditionally used parametric models have their own strengths compared to GAMs. The spline functions used in GAM can take unrealistic forms and lead to unexpected results especially if applied outside of the original model fitting data. Parametric GLM models offer more predictable model behaviour and provide easier implementation of models in new applications and more straightforward interpretation [14]. These differences support the use of GLM instead of GAM when the added flexibility of GAM does not provide clear improvements in model results.

The resolution of the snow damage probability map (16 x 16 m) was selected to match the resolution of the forest resource maps available for Finland [32], which were also used as input

data for our map. While the damage observations used in the model fitting were on stand level, we decided to use this resolution instead of a stand level polygon map to increase the usability of the map as it now matches the existing data and does not predefine the stand borders. However, we would recommend aggregating the map to stand level rather than interpreting single pixel values, which is also how the testing of the map was done here.

## Conclusions

Our results contribute to improving the scientific understanding of drivers of forest snow damage, which has so far been studied considerably less than, for example, wind disturbances. While the data used in the study covered only Finland, the results do provide insights into susceptibility of forests to snow disturbances also in other areas. However, it is possible that in conditions not covered in our data the factors driving snow damage may differ from our results and more research would be needed to apply our approach to areas with different species composition or high-altitude areas, which were not well represented in our data. The need for caution in application of results to new areas is highlighted by the geographical variation in the species sensitivity to snow damage in our results.

In this study, we demonstrated the applicability of the damage probability mapping approach for snow disturbances, using NFI data in combination with GIS data layers, and tested the performance of the resulting map. We also showed that including information about both abiotic factors and forest properties led to the best outcome in modelling snow damage probability, encouraging the combination of crown snow load data with local high-resolution and up-to date forest information. The developed statistical model can be used to assess snow damage probability of forests, either in specific snow damage events by using observed snow load data or more generally by using data of long-term snow load return-rates or projections of future snow loads.

## Supporting information

**S1 File. Details of model selection.**
(PDF)

## Acknowledgments

We would like to thank the National Forest Inventory of Finland for the NFI data we were able to use in the study. We acknowledge CSC–IT Center for Science, Finland, for computational resources.

## Author Contributions

**Conceptualization:** Susanne Suvanto, Aleksi Lehtonen, Seppo Nevalainen, Ilari Lehtonen, Heli Viiri, Mikko Peltoniemi.

**Data curation:** Susanne Suvanto, Seppo Nevalainen, Ilari Lehtonen, Mikael Strandström.

**Formal analysis:** Susanne Suvanto.

**Funding acquisition:** Heli Viiri, Mikko Peltoniemi.

**Investigation:** Susanne Suvanto.

**Methodology:** Susanne Suvanto, Ilari Lehtonen.

**Project administration:** Heli Viiri, Mikko Peltoniemi.

**Resources:** Mikael Strandström.

**Supervision:** Aleksi Lehtonen.

**Visualization:** Susanne Suvanto.

**Writing – original draft:** Susanne Suvanto.

**Writing – review & editing:** Susanne Suvanto, Aleksi Lehtonen, Seppo Nevalainen, Ilari Lehtonen, Heli Viiri, Mikael Strandström, Mikko Peltoniemi.

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
