## [Decision Letter · Decision Letter 0]

18 Apr 2021

PONE-D-21-06232

Mapping the probability of forest snow disturbances in Finland

PLOS ONE

Dear Dr. Suvanto,

Thank you for submitting your manuscript to PLOS ONE. After careful consideration, we feel that it has merit but does not fully meet PLOS ONE’s publication criteria as it currently stands. Therefore, we invite you to submit a revised version of the manuscript that addresses the points raised during the review process.

We look forward to receiving your revised manuscript.

Kind regards,

Sergio Rossi

Academic Editor

PLOS ONE

Journal Requirements:

We note that Figure 5 in your submission contain map images which may be copyrighted. All PLOS content is published under the Creative Commons Attribution License (CC BY 4.0), which means that the manuscript, images, and Supporting Information files will be freely available online, and any third party is permitted to access, download, copy, distribute, and use these materials in any way, even commercially, with proper attribution. For these reasons, we cannot publish previously copyrighted maps or satellite images created using proprietary data, such as Google software (Google Maps, Street View, and Earth). For more information, see our copyright guidelines: http://journals.plos.org/plosone/s/licenses-and-copyright.

2a, You may seek permission from the original copyright holder of Figure 5 to publish the content specifically under the CC BY 4.0 license. 

2b, If you are unable to obtain permission from the original copyright holder to publish these figures under the CC BY 4.0 license or if the copyright holder’s requirements are incompatible with the CC BY 4.0 license, please either i) remove the figure or ii) supply a replacement figure that complies with the CC BY 4.0 license. Please check copyright information on all replacement figures and update the figure caption with source information. If applicable, please specify in the figure caption text when a figure is similar but not identical to the original image and is therefore for illustrative purposes only.

Reviewers' comments:

Reviewer's Responses to Questions

**Comments to the Author**

1. Is the manuscript technically sound, and do the data support the conclusions?

Reviewer #1: Yes

Reviewer #2: Yes

2. Has the statistical analysis been performed appropriately and rigorously? 

Reviewer #1: Yes

Reviewer #2: Yes

3. Have the authors made all data underlying the findings in their manuscript fully available?

Reviewer #1: Yes

Reviewer #2: Yes

4. Is the manuscript presented in an intelligible fashion and written in standard English?

Reviewer #1: Yes

Reviewer #2: Yes

5. Review Comments to the Author

Reviewer #1: GENERAL COMMENTS

The present study concerns snow damages in Finnish forests and focuses on 1) improving the understanding of snow damage risks by taking into account abiotic and stand factors, 2) making a Finnish scale snow damage probability map.

Their study fits into the problematics of northern forest management and they took the time to describe every steps of their analysis. The results will help improve the general knowledge on snow damages on boreal/hemiboreal forests. This type of study is important for Nordic forestry because damage due to snowloads on crown can cause additionnal economic losses for forest managers but also on structures such as power lines and railways. Their study allows to go one step further in understanding and identifying the drivers of snow disturbances.

The topic was well introduced, the results and conclusion addresses the problematics, and the discussion addresses the limitations of the study.

As the study design and methodology seems robust, I recommend Minor revisions for this paper.

Therefore, I would like to congratulate the authors for their good (and cool) work and for providing a clean and complete manuscript.

DETAILED COMMENTS

First, thank you for advancing the knowledge in the field of snow damages on trees.

As a non-native English speaker, I will not judge the English language used in the manuscript, and I apologize for any potential misphrasing in this review.

Here are a few comments to help you improve your manuscript:

1) If you have the information, could you clarify a bit how was assessed the differences between wind damages and snow damages (NFI data)? I know that the study is about snow, but we know that wind and snow are strong co-agents in tree overturning or breakage. You assess this issue in your discussion L460 to 490, but I strongly suggest to further introduced it in the introduction or in the methods section as it is the main concern when studying wind/snow damages. It could be useful for out-of-the-scope readers.

2) L103-112 you detailed NFI data saying that you have data by blocks of 5 years, but you are able to plot yearly data in Figure 1? Could you elaborate a little bit more on the sampling method for people not acquainted with the NFI? Just to have in mind from the beginning that you have yearly data (on temporary plots).

3) Table 1: please add a note on the table to specify why you separated 2018 from the other years. It will allow the reader to see it quickly, as the information appears only later in the text.

4) L48: please add a reference for the economic losses.

5) L49: please add a reference concerning the powerlines.

6) L64: “snow disturbances play an important role”: in what? Please specify.

7) L70: remove “typically” as it appears at the beginning of the sentence.

8) L87: add “snow” in front of damage.

9) L150-151: please rephrase this sentence

10) L244: please correct “(A)” (the parenthesis is after “year” L246 instead of after the “A”.)

11) L261-262: please merge the two sentences and add a reference to the R version you are using

12) L312: please add units to your variables inside the table

13) FIGURE 1: please add the units after “snowload”.

14) FIGURE 2: please add the units after each quantitative variable (Snowload, DBH,…)

15) FIGURE 3: please add the units after each quantitative variable (Snowload, DBH,…)

16) FIGURE 5: this figure is low resolution (pixelized) please switch it with a higher resolution version.

17) Figures: I do not know which color palette you used. If it is not already the case, I strongly recommend using colorblind friendly colors for your figures (a lot of colorblind palette are available on R).

Reviewer #2: This is a well written and clear paper that deserves to be published. We know relatively little about snow damage compared to, for example, wind damage, and it is difficult to predict because of the specific meteorological conditions required. Therefore, any paper that advances our knowledge of the factors affecting snow damage is important, especially because the locations of snow damage is likely to change in a changing climate.

There are a few points that need attention. A number of these are in the annotated version of your paper (please see attached file) and the main ones are given below:

1. There is no definition of differences in "Damage Severity" as discussed in Table 1. Is this information ever used in the paper?

2. What is the difference between "broken trees" and "stem damage" in Table 1.

3. There is very little discussion about correlations between input variables, e.g. between dbh and basal area and how you dealt with this in the model development.

4. There was no clear justification for the input variables chosen (lines 132-138). They make sense to me but you should expand this section to say why they might be important.

5. For the stands with trees with dominant height less than 1.3 m (lines 140-141) and dbh was set to 0 cm is this a problem. Were these stands excluded from the modelling?

6. Could you not use stand dominant height in the modelling (line 140)? You do not seem to have any height data in the modelling.

7. It was not clear why you chose a spatial modelling resolution of 16x16 m. What is your justification? I cannot see how anyone would use such high resolution data when the forest data is at stand level. I always worry about models that are at spatial resolutions much higher than the input data. It can give a false sense of precision.

7. Lines 275-282. How is this data from the Forest Centre obtained. It is very unclear. Is it from declarations made by private forest owners?

8. I am not convinced by the use of GAM models. They do not seem to do a better job except in one situation. Also the uncertainty in the model predictions seems higher and the shape of the model responses (Figure 3) do not look realistic. If a non-linear model is not doing a substantially better job than a linear model we should prefer the linear model. They are more easily used in an area other than were the model was developed and also they are less likely to behave in a strange way when pushed outside the envelope of data that was used to create them. I think you need to justify more the use of the GAM models.

9. I would like to have seen in the Discussion something about how the models you have developed might be applied or adapted to other countries with snow damage to forests, e.g. Sweden, Norway, Estonia, Russia. Otherwise the paper can appear too Finland focused, but we know snow damage is moving to new areas with changing climate and foresters need to be able to predict the possibility of snow damage in their own countries.

6. PLOS authors have the option to publish the peer review history of their article (what does this mean?). If published, this will include your full peer review and any attached files.

Reviewer #1: **Yes: **Marine Duperat

Reviewer #2: **Yes: **Barry Gardiner

---

## [Author Response · Author response to Decision Letter 0]

22 May 2021

We would like to thank the reviewers for their kind comments and constructive suggestions on our work. We have now revised the manuscript following their feedback and believe that this has significantly improved the quality of the manuscript. 

Please find our point-to-point responses to each of the comments below, these are also submitted as a separate file which may be easier to read (different colors for our replies).

On behalf of the author team,

Susanne Suvanto

Response to reviewers

Suvanto et al. Mapping the probability of forest snow disturbances in Finland 

We note that Figure 5 in your submission contain map images which may be copyrighted. 

The background map in the figure is from Natural Earth (public domain data) and the Forest Centre data in the first map is published under CC BY 4.0 licence (as described in the Methods section). Other data in the maps is results calculated in this study. Added info about background map and the Forest Centre data licence to the caption. 

Reviewer #1: GENERAL COMMENTS

The present study concerns snow damages in Finnish forests and focuses on 1) improving the understanding of snow damage risks by taking into account abiotic and stand factors, 2) making a Finnish scale snow damage probability map.

Their study fits into the problematics of northern forest management and they took the time to describe every steps of their analysis. The results will help improve the general knowledge on snow damages on boreal/hemiboreal forests. This type of study is important for Nordic forestry because damage due to snowloads on crown can cause additionnal economic losses for forest managers but also on structures such as power lines and railways. Their study allows to go one step further in understanding and identifying the drivers of snow disturbances.

The topic was well introduced, the results and conclusion addresses the problematics, and the discussion addresses the limitations of the study.

As the study design and methodology seems robust, I recommend Minor revisions for this paper.

Therefore, I would like to congratulate the authors for their good (and cool) work and for providing a clean and complete manuscript.

DETAILED COMMENTS

First, thank you for advancing the knowledge in the field of snow damages on trees.

As a non-native English speaker, I will not judge the English language used in the manuscript, and I apologize for any potential misphrasing in this review.

Here are a few comments to help you improve your manuscript:

1) If you have the information, could you clarify a bit how was assessed the differences between wind damages and snow damages (NFI data)? I know that the study is about snow, but we know that wind and snow are strong co-agents in tree overturning or breakage. You assess this issue in your discussion L460 to 490, but I strongly suggest to further introduced it in the introduction or in the methods section as it is the main concern when studying wind/snow damages. It could be useful for out-of-the-scope readers.

Added clarification to text: “The definition of the damage agent and the estimation of time since damage are based on the judgement of the NFI field team on site.”

2) L103-112 you detailed NFI data saying that you have data by blocks of 5 years, but you are able to plot yearly data in Figure 1? Could you elaborate a little bit more on the sampling method for people not acquainted with the NFI? Just to have in mind from the beginning that you have yearly data (on temporary plots).

The x-axis shows the year of the NFI field work on the plot, but the damage may have occurred within 5 years from the field visit. We have now clarified this in the figure labels.

3) Table 1: please add a note on the table to specify why you separated 2018 from the other years. It will allow the reader to see it quickly, as the information appears only later in the text.

Added explanation to the caption: “The values are also shown separately for 2005-2017 with typical snow conditions and year 2018 with exceptionally high snow loads.”

4) L48: please add a reference for the economic losses.

Added reference (Nykänen et al. 1997)

5) L49: please add a reference concerning the powerlines.

Added reference (Groenemeijer et al. 2015)

6) L64: “snow disturbances play an important role”: in what? Please specify.

Specified the sentence: “In the disturbance dynamics of northern forest ecosystems, snow disturbances play an important role”

7) L70: remove “typically” as it appears at the beginning of the sentence.

Changed as suggested

8) L87: add “snow” in front of damage.

Changed as suggested

9) L150-151: please rephrase this sentence

Rephrased the sentence to improve clarity.

10) L244: please correct “(A)” (the parenthesis is after “year” L246 instead of after the “A”.)

Changed as suggested

11) L261-262: please merge the two sentences and add a reference to the R version you are using

Changed as suggested

12) L312: please add units to your variables inside the table

The units and full descriptions of the model covariates are in Table 2, I have not added them here to make the table easier to read (e.g. for interaction terms and log transformed variables the readability would suffer from inserting extra information after each covariate name). I have now added a note to the caption to make the unit information easier to find for the reader: “See full descriptions and units of the covariates in Table 2.”

13) FIGURE 1: please add the units after “snowload”.

Changed as suggested

14) FIGURE 2: please add the units after each quantitative variable (Snowload, DBH,…)

Changed as suggested

15) FIGURE 3: please add the units after each quantitative variable (Snowload, DBH,…)

Changed as suggested

16) FIGURE 5: this figure is low resolution (pixelized) please switch it with a higher resolution version.

To my understanding the figure resolution is lower in the review version. We have now checked that the original resolution is indeed better and will make sure to pay special attention in the proof stage that the resolution of figures is sufficient after type setting.

17) Figures: I do not know which color palette you used. If it is not already the case, I strongly recommend using colorblind friendly colors for your figures (a lot of colorblind palette are available on R).

Thank you for bringing this up, we have now changed the color palettes in figures 4 and where the original colors were not colorblind friendly (with red and green with similar intensity) and checked the rest of the figures with the “Coblis —Color Blindness Simulator” tool (https://www.color-blindness.com/coblis-color-blindness-simulator/).

Reviewer #2: This is a well written and clear paper that deserves to be published. We know relatively little about snow damage compared to, for example, wind damage, and it is difficult to predict because of the specific meteorological conditions required. Therefore, any paper that advances our knowledge of the factors affecting snow damage is important, especially because the locations of snow damage is likely to change in a changing climate.

There are a few points that need attention. A number of these are in the annotated version of your paper (please see attached file) and the main ones are given below:

1. There is no definition of differences in "Damage Severity" as discussed in Table 1. Is this information ever used in the paper?

We have added descriptions of each damage severity class into a footnote in Table 1.

The information about damage severity or type is not specifically used in the analysis as all the snow damage cases are grouped together. This information in Table 1 is still presented to give the reader a clear idea of what kind of damage the ‘snow damage’ in our analysis actually is. We have revised the text to make this clearer: 

“In our analysis we grouped all the snow damage observations together and therefore the analysis does not consider differences in damage type or severity. Table 1 shows the distribution of the damage observations in the different damage type and severity classes.”

2. What is the difference between "broken trees" and "stem damage" in Table 1.

‘Broken trees’ refers to stem breakage and ‘stem damage’ to all other stem damage types. We have not clarified this difference in the table.

3. There is very little discussion about correlations between input variables, e.g. between dbh and basal area and how you dealt with this in the model development.

In the model selection the multicollinearity of covariates was considered with the variance inflation factor. We do agree that more information about the correlations between model covariates would still be useful especially from the point of view of interpreting the model coefficients. Therefore, we have added a correlation matrix of the continuous predictor variables in the final model (new Fig. 1) and added discussion about how the correlated covariates affect the interpretation of model coefficients: 

“In the interpretation of model results it is important to consider the correlations between the model predictors. For example, if considering forest stands with the same DBH, the stands with higher basal area have a higher snow damage probability (as illustrated in Figs 2 and 3). However, these two variables are positively correlated, meaning that an increase of basal area is usually accompanied with an increase in DBH, which in turn has a negative effect on the damage probability. Therefore, the higher basal area leads to increased damage probability only if the change in DBH is not large enough to counter this effect. Similarly, snow load, long term snow load and altitude are correlated, and their effects should be considered together when interpreting the model coefficient estimates.”

4. There was no clear justification for the input variables chosen (lines 132-138). They make sense to me but you should expand this section to say why they might be important.

We have now improved the text about model predictors and their selection. In the previous version the description of (potential) model predictors was divided in several parts – first in descriptions of different data sets and then second time under the “Statistical modelling” subtitle. Now we concentrated the description of model selection and all the variables considered in the selection process in the “Statistical modelling” section and hope that this makes the process easier to understand for the reader.

5. For the stands with trees with dominant height less than 1.3 m (lines 140-141) and dbh was set to 0 cm is this a problem. Were these stands excluded from the modelling?

Stand average DBH for these stands was set to 0 since trees (seedlings) with height lower than 1.3 meters do not have a diameter at breast height (1.3 meters). However, they were not excluded from the model, they just have value of 0 cm for DBH. We have revised the text to make this more clear: “For these stands the stand average DBH was set to 0 cm, as the seedling do not reach the breast heigh (1.3 m).”

6. Could you not use stand dominant height in the modelling (line 140)? You do not seem to have any height data in the modelling.

Height data was considered in the model selection process, but it was very strongly correlated with the DBH and the model comparison led to selection of DBH for the final model instead of height. Following from a comment in the pdf, we noticed a rather embarrassing mistake of an extra ‘0’ in the p-value threshold (0.001 instead of 0.01), this is now fixed and double checked that the actual model selection process did not exclude variables with unnecessarily low threshold.

We have now modified the text accordingly:

“In the model selection process variables were left out of the model based on three reasons: (1) high p-values (p > 0.01) or no improvement in AIC, (2) high variance inflation factors, suggesting high multicollinearity between variables and (3) a lack of national level wall-to-wall spatial data available for the creation of the snow damage probability map. Potential predictors excluded from the final model based on p-values and AIC values were variables describing thinning and precommercial thinning in the stand, site type, number of species and percentage of basal area covered by the dominant species. High GVIF affected the model selection when both DBH and average tree height were included in the model. To decide which one of these two would be included in the final model, we compared AIC values of models with DBH but no height and with height but no DBH. The model with DBH showed lower AIC value and therefore DBH was selected for the final model instead of tree height. Lack of available spatial data led to exclusion of two variables from the model that could have been included based on the other model selection criteria. These were variables describing tending of seedling stands (negative coefficient estimate, i.e., lowering the damage probability, p = 0.002) and Shannon diversity index of tree species, calculated from the species-specific share of basal area (negative coefficient estimate, i.e. lowering the damage probability, p = 0.004). Full results for models with variables not included in the final model are found in supplementary material (S1).”

7. It was not clear why you chose a spatial modelling resolution of 16x16 m. What is your justification? I cannot see how anyone would use such high resolution data when the forest data is at stand level. I always worry about models that are at spatial resolutions much higher than the input data. It can give a false sense of precision.

We have now clarified the reasoning behind the resolution in the text: 

“The resolution of the snow damage probability map (16 x 16 m) was selected to match the resolution of the forest resource map available for Finland [28], which was also used as an input of our map. While the damage observations used in the model fitting were on stand level, we decided to use this resolution instead of a stand level polygon map to increase the usability of the map as it matches the existing data and does not predefine the stand borders. However, for best accuracy we would recommend aggregating the map to stand level rather than interpreting single pixel values, which is also how the testing of the map was done here.”

7. Lines 275-282. How is this data from the Forest Centre obtained. It is very unclear. Is it from declarations made by private forest owners?

Added clarification to text: 

“This database is based on a combination of data from different sources, such as remote sensing, field measurement and reports from forest owners”

8. I am not convinced by the use of GAM models. They do not seem to do a better job except in one situation. Also the uncertainty in the model predictions seems higher and the shape of the model responses (Figure 3) do not look realistic. If a non-linear model is not doing a substantially better job than a linear model we should prefer the linear model. They are more easily used in an area other than were the model was developed and also they are less likely to behave in a strange way when pushed outside the envelope of data that was used to create them. I think you need to justify more the use of the GAM models.

We do agree with the reviewer here that since the results for two methods are so similar, it is better to stick with GLMs. The purpose of testing the two methods was to see if the non-parametric splines would do better in modelling potential non-linear responses. However, from the results one can see that there is no clear improvement. We think that this is interesting on its own right and justifies the presentation of the GAM model results along with the GLM results. We have modified the discussion of the GLM vs GAM methods to make these points clearer:

“While the flexible smoothing spline functions in GAM increased the model performance in the case of the abiotic predictors, the traditionally used parametric models have their own strengths compared to GAMs. The spline functions used in GAM can take unrealistic forms and lead to unexpected results especially if applied outside of the original model fitting data. Parametric GLM models offer more predictable model behaviour and provide easier implementation of models in new applications and more straightforward interpretation [13]. These differences support the use of GLM instead of GAM when the added flexibility of GAM does not provide clear improvements in model results.”

9. I would like to have seen in the Discussion something about how the models you have developed might be applied or adapted to other countries with snow damage to forests, e.g. Sweden, Norway, Estonia, Russia. Otherwise the paper can appear too Finland focused, but we know snow damage is moving to new areas with changing climate and foresters need to be able to predict the possibility of snow damage in their own countries.

We have added discussion on the applicability of the results outside of Finland:

“Our results contribute to improving the scientific understanding of drivers of forest snow damage, which has so far been studies considerably less than, for example, wind disturbances. While the data used in the study covered only Finland, the results do provide insights into susceptibility of forests to snow disturbances also in other areas. However, it is possible that in conditions not covered in our data the factors driving snow damage may differ from our results and more research would be needed to apply our approach to areas with different species composition or high-altitude areas, which are not well represented in our data. The need for caution in application of results to new areas is highlighted by the geographical variation in the species sensitivity to snow damage in our results.”

---

## [Editor Report · Decision Letter 1]

28 May 2021

PONE-D-21-06232R1

Mapping the probability of forest snow disturbances in Finland

PLOS ONE

Dear Dr. Suvanto,

Thank you for submitting your manuscript to PLOS ONE. After careful consideration, we feel that it has merit but does not fully meet PLOS ONE’s publication criteria as it currently stands. Therefore, we invite you to submit a revised version of the manuscript that addresses the points raised during the review process.

Thanks for your revisions. Before completing the evaluation process, I have few minor comments to improve the figures:

-Fig 2: the x-axis is in common, thus the figure should involve two plots in the same column (at the top and at the bottom), and not one on the left and one on the right. One horizontal axis should be present

-Figs 3 and 4: add a clear vertical label (“Fitted” is meaningless for readers). Do not repeat labels and values of the vertical axes within the same row. Add also minor or major marks on the axes

-The range of AUC is 0.55-0.80, keep this range for the y-axis, thus reducing the empty space

We look forward to receiving your revised manuscript.

Kind regards,

Sergio Rossi

Academic Editor

PLOS ONE
---

## [Author Response · Author response to Decision Letter 1]

29 Jun 2021

POINT-TO-POINT RESPONSES TO COMMENTS

- Fig 2: the x-axis is in common, thus the figure should involve two plots in the same column (at the top and at the bottom), and not one on the left and one on the right. One horizontal axis should be present

OUR RESPONSE: changed as suggested

- Figs 3 and 4: add a clear vertical label (“Fitted” is meaningless for readers). Do not repeat labels and values of the vertical axes within the same row. Add also minor or major marks on the axes

OUR RESPONSE: changed as suggested

- The range of AUC is 0.55-0.80, keep this range for the y-axis, thus reducing the empty space

OUR RESPONSE: changed as suggested

---

## [Editor Report · Decision Letter 2]

6 Jul 2021

Mapping the probability of forest snow disturbances in Finland

PONE-D-21-06232R2

Dear Dr. Suvanto,

We’re pleased to inform you that your manuscript has been judged scientifically suitable for publication and will be formally accepted for publication once it meets all outstanding technical requirements.

Kind regards,

Sergio Rossi

Academic Editor

PLOS ONE
---

## [Editor Report · Acceptance letter]

21 Jul 2021

PONE-D-21-06232R2 

Mapping the probability of forest snow disturbances in Finland 

Dear Dr. Suvanto:

I'm pleased to inform you that your manuscript has been deemed suitable for publication in PLOS ONE. Congratulations! Your manuscript is now with our production department. 

Kind regards, 

on behalf of

Prof. Sergio Rossi 

Academic Editor

PLOS ONE